# COVID-19 vaccine brand hesitancy and other challenges to vaccination in the Philippines

**Arianna Maever L. Amit**[1]*, **Veincent Christian F. Pepito**[1], **Lourdes Sumpaico-Tanchanco**[1,2], **Manuel M. Dayrit**[1]

1 School of Medicine and Public Health, Ateneo de Manila University, Manila, Philippines, 2 The Medical City, Manila, Philippines

* aamit@ateneo.edu

**Data Availability Statement:** All data relevant to the study are included in the article.

**Funding:** AMLA/VCFP/LST/MMD are funded by the Ateneo de Manila University Research Council

## Abstract

Effective and safe COVID-19 vaccines have been developed at a rapid and unprecedented pace to control the spread of the virus, and prevent hospitalisations and deaths. However, COVID-19 vaccine uptake is challenged by vaccine hesitancy and anti-vaccination sentiments, a global shortage of vaccine supply, and inequitable vaccine distribution especially among low- and middle-income countries including the Philippines. In this paper, we explored vaccination narratives and challenges experienced and observed by Filipinos during the early vaccination period. We interviewed 35 individuals from a subsample of 1,599 survey respondents 18 years and older in the Philippines. The interviews were conducted in Filipino, Cebuano, and/or English via online platforms such as Zoom or via phone call. All interviews were recorded, transcribed verbatim, translated, and analysed using inductive content analysis. To highlight the complex reasons for delaying and/or refusing COVID-19 vaccines, we embedded our findings within the social ecological model. Our analysis showed that individual perceptions play a major role in the decision to vaccinate. Such perceptions are shaped by exposure to (mis)information amplified by the media, the community, and the health system. Social networks may either positively or negatively impact vaccination uptake, depending on their views on vaccines. Political issues contribute to vaccine brand hesitancy, resulting in vaccination delays and refusals. Perceptions about the inefficiency and inflexibility of the system also create additional barriers to the vaccine rollout in the country, especially among vulnerable and marginalised groups. Recognising and addressing concerns at all levels are needed to improve COVID-19 vaccination uptake and reach. Strengthening health literacy is a critical tool to combat misinformation that undermines vaccine confidence. Vaccination systems must also consider the needs of marginalised and vulnerable groups to ensure their access to vaccines. In all these efforts to improve vaccine uptake, governments will need to engage with communities to 'co-create' solutions.

COVID-19 Research Grant (Grant No. COVID-URC 01 2021). The funder had no role in study design, data collection and analysis, decision to publish, or preparation of the manuscript.

**Competing interests:** We have read the journal's policy and the authors of this manuscript have the following competing interests: VCFP owns shares of GMA Network, Inc., a Philippine Stock Exchange-listed company with interests in mass media. AMLA, VCFP, and MMD receive funding from Sanofi to conduct research on self-care.

## Introduction

The coronavirus disease 2019 (COVID-19) pandemic continues to burden health systems and communities globally, with millions of cases and deaths [1]. Because of the significant and continued impact of COVID-19, vaccines have been developed at a rapid and unprecedented pace to control the spread of the virus, and prevent hospitalisations and deaths [2]. Many vaccines have been shown to be safe and effective with high-income countries having vaccinated more than half of their population [3]. Despite the availability of these vaccines, countries are faced with various challenges including vaccine hesitancy and anti-vaccination sentiments, limited global supply, and inefficient vaccine deployment [4, 5]. These issues in vaccine uptake, together with declining community acceptance of other public health interventions, will mean a delayed recovery and prolonged pandemic [6].

The World Health Organization (WHO) in 2019 identified vaccine hesitancy or the reluctance to vaccinate as one of the top ten threats to global health despite evidence of the important role of vaccines in improving population health outcomes [7]. Together with weak primary health care and other health challenges, countries especially low- and middle-income countries (LMICs) will struggle to meet the demands of the communities within their health system capacity. With the pandemic, countries are further burdened with many health systems overwhelmed throughout its course. The Philippines presently faces these challenges: vaccine hesitancy and increasing anti-vaccination sentiments, a weak primary health care system with efforts to strengthen it through the recently implemented Universal Health Care Law, and an overwhelmed health system because of the demands of COVID-19 and other public health problems [8–13]. These challenges are further compounded by a global shortage of vaccine supply with inequitable vaccine distributions [14].

Historically, the Philippines was one of the countries with generally high vaccine confidence rates [15]. Following the dengue vaccine controversy in 2017 however, confidence levels have dramatically dropped and have impacted succeeding vaccination efforts including the COVID-19 vaccination campaign [9, 12, 15–17]. Dengvaxia, the world's first commercially available dengue vaccine developed by Sanofi Pasteur, was introduced as part of a national school-based immunization programme despite the lack of empirical data on the risks associated with administration of the vaccine among those not previously infected with dengue or seronegative children [9, 12, 15–17]. By the time reports were released that the vaccine may cause more severe disease among seronegatives, the Philippines had already inoculated more than 800,000 Filipino school-age children [9]. This was highly politicised, and damaged trust in vaccines and the health sector [9, 12, 15–17]. As a result, immunisation rates dropped and the country saw outbreaks of previously controlled vaccine-preventable diseases such as measles and polio [18, 19]. In addition to vaccine hesitancy, the Philippine health system is not prepared for additional health care demands. As early as the first phase of the pandemic, critical care capacity was overwhelmed with the influx of patients in hospitals [10, 11]. As of 16 September 2021, the Philippines ranks third among countries with the highest number of newly confirmed cases per one million population [1, 20]. Globally, 42.9% of the world population have received one dose of a COVID-19 vaccine, with much lower rates in LMICs like the Philippines [20, 21]. Only 55% of Filipinos have expressed willingness to be vaccinated against COVID-19, and as of 16 September 2021, only 30% of the population have been fully vaccinated[21, 22].

To end this pandemic, it is critical to implement all possible public health interventions and strategies from face masks, physical distancing, to getting vaccinated [4, 23]. However, there is a need to recognise that the adoption of all these interventions is influenced by individual risk perceptions, and these perceptions are shaped by various sources of information and experiences [24]. Additionally, there are interpersonal and structural factors that influence health

decisions of individuals. Recognising the multiple dimensions in which behaviours and decisions occur, theories and models have been proposed to explain how individuals make decisions on their health based on factors that change over time and context [25, 26]. The social ecological model provides a useful framework for investigating health behaviours and decisions by recognising that a multiplicity of factors interacts to influence health of individuals [26]. These include individual factors representing biological or behavioural characteristics, interpersonal factors representing networks and social capital operating within a defined boundary, and structural factors that include health systems and are mediated through laws and policies [26]. Published studies on vaccination that utilised this model reported that vaccine intentions and attitudes operate along multiple dimensions, with a series of events influencing decisions related to vaccination [17, 27, 28]. Improving adherence to interventions and vaccination rates therefore requires a better understanding of the different reasons behind vaccine mistrust and not just determining their individual beliefs, knowledge, and levels of trust [17, 27, 28]. A recently published scoping review supports the use of the social ecological model in understanding attitudes towards COVID-19 vaccination [29]. The review showed that influencing factors are embedded within the social ecological model and that multilevel interventions are needed to improve uptake of vaccines [29]. This scoping review of 50 articles had representation from various countries, but did not include data from the Philippines. We address this gap by exploring the vaccination narratives and challenges experienced and observed by Filipinos during the early COVID-19 vaccination period. We used qualitative data from a mixed-methods study conducted from June to August 2021 that aimed to understand how people in the Philippines view COVID-19 and what influences their behaviours. With these findings, we hope to provide insights to possible avenues of future research and directions for improving COVID-19 vaccine uptake and reach.

## Material and methods

### Design and setting

We conducted an online survey among adults ages 18 and older in the Philippines (n = 1,599) from June to August 2021. A subsample participated in the semi-structured interviews (n = 35) with representation from the general population and health workforce from July to August 2021. Data from the interviews informed the findings of this paper.

### Participants and recruitment

We aimed to interview participants from different regions in the Philippines, various age groups, socio-economic classes, and vaccination status and attitudes. This allowed us to ensure maximum variation sampling, which aims to capture as many population contexts as possible. We contacted a total of 115 individuals through the information they provided (i.e., mobile number, phone number, e-mail). Out of the 115, 35 participants completed the interviews. The remaining 80 either refused or could not be contacted after a maximum of three attempts. We classified participants according to their vaccination priority group based on the COVID-19 Vaccination Program's prioritisation framework [30]. Those in the first priority group (A1) were frontline workers in health facilities; other priority groups (A2 to C) comprised and represented the general population (**Table 1**).

### Data collection

We conducted the interviews in Filipino, Cebuano, and/or English via online platforms such as Zoom or via phone call. The interview guide included questions about their views on

**Table 1. Vaccine prioritisation framework (2020).**

| Priority group | Eligible individuals |
|---|---|
| A1 | Workers in frontline health services |
| A2 | All senior citizens |
| A3 | Persons with comorbidities |
| A4 | Frontline personnel in essential sectors, including uniformed personnel |
| A5 | Indigent population |
| B1 | Teachers, social workers |
| B2 | Other government workers |
| B3 | Other essential workers |
| B4 | Socio-demographic groups at significantly higher risk other than senior citizens and poor population |
| B5 | Overseas Filipino Workers (OFWs) |
| B6 | Other remaining workforce |
| C | Rest of the Filipino population not otherwise included in the above groups |

COVID-19, vaccines, and their risk perceptions and behaviours. We recruited interview participants until saturation was reached (i.e., no new information was being obtained from the interviews) [31]. The interviews lasted between 60 to 90 minutes with a token amounting to USD 6 provided to each participant. All participants consented to the interview being recorded.

## Data analysis

The interviews were digitally recorded, transcribed verbatim, and translated from Filipino or Cebuano to English. The research team are native and/or fluent speakers of the three languages, and checked for linguistic and conceptual equivalence in the translated documents. We de-identified all participants and assigned pseudonyms. We analysed the data using inductive content analysis focusing on the experiences and views towards vaccination [32]. Our analysis was guided by principles of grounded theory. Transcripts of the interviews were read to identify themes and two investigators (AMLA, VCFP) independently coded the interviews according to emergent themes in Microsoft Excel [33]. We used coding language that was close to the participants' terms and phrases to ensure that we were co-constructing accurate categories reflective of their responses [34]. The codes were reviewed, and areas of disagreement were resolved between the two investigators. Themes from the interviews were further explored through discussions with the other members of the team. We considered reflexivity throughout data collection and analysis, acknowledging that our preconceptions and experiences about vaccination as public health practitioners and health professionals may influence the way we analyse and interpret data. Our use of the grounded theory allowed us to explore the experiences of our participants and our own shared experiences, and avoided being limited by how we view COVID-19 vaccination [35]. To highlight the complex reasons for delaying and/or refusing COVID-19 vaccination, we embedded our findings within the social ecological model with three broad themes: individual factors (attitudes, beliefs, knowledge, behaviours), interpersonal factors (relationships and social networks), and structural factors (health systems and service delivery; media; and policies, regulations, and laws at the local, national, and global level) [26] (**Fig 1**). The quotes presented in this paper are either in the original English or translated from Filipino or Cebuano.

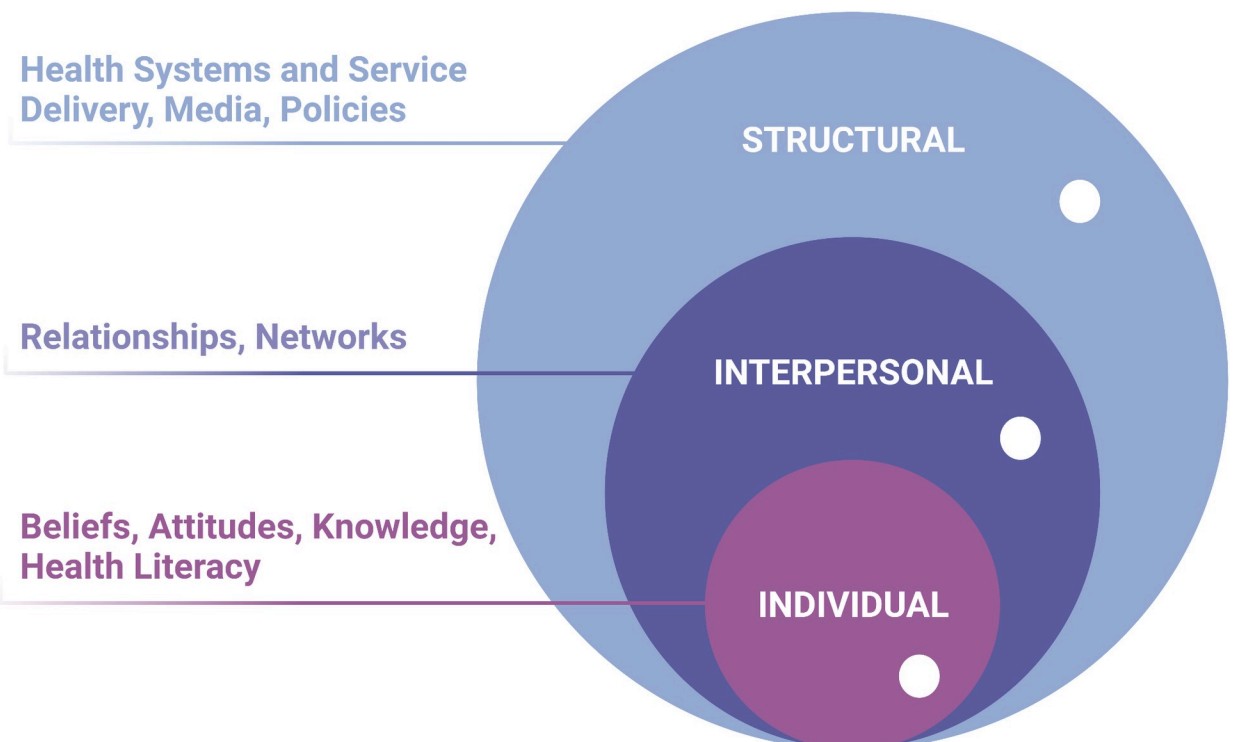

**Fig 1. Social ecological model applied to COVID-19 vaccination.** This figure shows the three main tiers of factors influencing vaccination intention and uptake: individual (beliefs, attitudes, knowledge, health literacy), interpersonal (relationships, networks), and structural (health systems and service delivery, media, policies). These three dimensions are jointly or individually impacted by misinformation (white circles).

### Patients and public involvement

The public were not directly involved in the design, recruitment, conduct, reporting, or dissemination plans of this research. Their only involvement was as research participants.

### Ethics statement

This study was approved by the University Research Ethics Office of Ateneo de Manila University (Study No. SMPH CORISK 2021). All participants were informed about the aims and objectives of the study by including the written consent form in the email correspondence. Prior the interview, the research team thoroughly explained the study to them and provided them the opportunity to ask questions they may have. Written digital consent was taken from study participants before the interview.

### Results

We interviewed 35 participants with representation from different vaccination priority groups working in various parts of the country. Our participants also had different educational backgrounds, employment status, and vaccination attitude (**Table 2**). There was an almost equal proportion of females and males (females: 19; males: 16) with a median age of 38 years old (range: 21 to 74 years old) in the overall study population.

Participant views on the barriers to COVID-19 vaccination are presented below, organised using the three tiers of the social ecological model. Individual barriers include perceptions; attitudes; and beliefs about the science, about vaccines, about the health system and

**Table 2. Characteristics of interview participants.**

| Characteristic | Priority group | | | | | |
|---|---|---|---|---|---|---|
| | A1 (n = 15) | A2 (n = 5) | A3 (n = 4) | A4 (n = 5) | B (n = 4) | C (n = 2) |
| | N (%) | N (%) | N (%) | N (%) | N (%) | N (%) |
| Median age (years) | 37 | 67 | 45 | 29 | 33 | 37 |
| Female | 9 (60) | 3 (60) | 2 (50) | 2 (40) | 3 (75) | 0 (0) |
| Married | 6 (40) | 4 (80) | 1 (25) | 1 (20) | 2 (50) | 1 (50) |
| Finished university education or higher | 15 (100) | 5 (100) | 4 (100) | 3 (75) | 3 (75) | 2 (100) |
| Positive towards COVID-19 vaccines | 14 (93) | 2 (40) | 2 (50) | 5 (100) | 2 (50) | 2 (100) |

government. Interpersonal barriers are the networks and social capital that influence health beliefs and decisions. Vaccine procurement, supply, and logistics, together with media- and policy-related issues, comprise the structural barriers. Where there are differences between the general population and health workers, these are highlighted in the text.

### Individual barriers

**Vaccine brand hesitancy and brand preferences.** Vaccine brand hesitancy or delay in getting the vaccine due to brand preferences was a common theme among the participants. The country's first administered vaccine was Sinovac-CoronaVac, which is manufactured by a Chinese biopharmaceutical company. This was given to health workers despite lack of published data on effectiveness at the time and initial announcements that these were not recommended for high-risk individuals (**Quote I1, Table 3**). In addition to concerns about the effectiveness of the vaccine, participants also read and heard information on how this vaccine was made. They believed this specific vaccine was using the same virus to 'immunise' an individual's system, which may have unintended effects (**Quote I2, Table 3**). Other participants cited that this specific brand was not recognised by other countries, and therefore wanted and waited for other vaccines. Meanwhile, others refused to receive mRNA vaccines due to beliefs about its safety and effectiveness.

**Negative experiences with the health system as source of vaccine hesitancy and anti-vaccination sentiments.** The participants cited negative experiences in the past, whether these happened recently or decades ago, as causes of their negative attitude towards vaccines. Three participants who identified themselves as COVID-19 'anti-vaxxers' or those opposed to vaccines, had different sources of anti-vaccination sentiments. These three participants belong to different priority groups. One belongs to the A1 or frontliner group and is working as a Barangay Health/Emergency Response Team (BHERT) member who responds to COVID-19 related health care needs in the community. The second is a retired professional (A2 or senior citizen group) while the third is an environmental protection officer who oversees implementation of public health standards in the community (B2 or other government workers). These participants experienced an undesired event related to vaccines and/or medical care from four years to more than three decades prior the pandemic (**Quotes I3-I5, Table 3**). Except for one anti-vaxxer, no other health worker reported negative experiences that caused mistrust in the COVID-19 vaccines and vaccination campaign.

**Vaccines are viewed as unsafe and deadly.** Perceptions on risk of getting infection with and dying from the virus varied among the participants. However, for those who were opposed to the vaccines, their fear of the COVID-19 vaccine and its effects was greater than their fear of the virus and outcomes (**Quote I6, Table 3**). This fear and their view of vaccines being unsafe and deadly resulted to vaccine refusals or delays. According to them, the deaths observed after

**Table 3. Illustrative quotes for individual barriers.**

| Quote ID | Theme | Illustrative quote |
|---|---|---|
| I1 | Vaccine brand hesitancy and brand preferences | I had concerns with Sinovac. I read about the studies published about vaccines and Sinovac initially did not publish their results. For me, I did not want Sinovac. [. . .] There was even a time when the Department of Health said, "Sinovac is not for health workers". I still think some vaccines are better than others (A1, 51–60 years old, male, Laguna). |
| I2 | Vaccine brand hesitancy and brand preferences | I had myself registered, then I did not go, then I registered, then did not go. Then when I went through the third registration, they [local government unit] asked in their website, "Why did you not show up the last time you were scheduled"? The choices for refusal were: conflict in schedule, choice of vaccine, and three other reasons. I chose choice of vaccine. After that, they scheduled me to another vaccination site to get my vaccine. When I went, I was still not in any way sure that it was Pfizer. But I knew it would not be Sinovac. So I was open to whatever vaccine it will be, except for Sinovac. So knowing that it was not Sinovac convinced me to go. Well, regardless if it was AstraZeneca or other vaccines, I was okay with it as long as it was not Sinovac. Because I heard that they used the same virus, they cased it in the vaccine, and it was like you will have the virus in your body? So the negativity as well as the news about it. That was really a major reason that I wouldn't go for Sinovac (A3, 41–50 years old, female, National Capital Region). |
| I3 | Negative experiences with the health system as source of vaccine hesitancy and anti-vaccination sentiments | I self-study medicine because I do not want to depend on doctors. If I did, maybe I died a long time ago. [. . .] Once upon a time in my life, I was admitted in the hospital. I was later diagnosed with UTI [urinary tract infection] but was initially asked for an ultrasound because they thought I was pregnant. Another doctor asked me to drink beer for my UTI. From that [day] on, I just self-studied medicine. With COVID, I don't really believe [what I read and hear] without validation. A former scientist working at Pfizer insisted that vaccines do not help COVID-19. Because vaccines are not good for our body. [. . .] And even the medical practitioners or doctors themselves cannot help if someone suffers the adverse reaction of the vaccine (A1, 51–60 years old, female, Misamis Oriental). |
| I4 | Negative experiences with the health system as source of vaccine hesitancy and anti-vaccination sentiments | I was working in a school and was looking after a child whose mother was busy. The child was very smart, but was diagnosed with autism. When he had check-ups, I would accompany him and observe. Then I researched about autism, and found that it is caused by vaccines. No wonder there are many children with autism; it is associated with vaccines (A2, 61–70 years old, female, Camarines Norte). |

(*Continued*)

**Table 3.** (Continued)

| Quote ID | Theme | Illustrative quote |
|---|---|---|
| I5 | Negative experiences with the health system as source of vaccine hesitancy and anti-vaccination sentiments | When my child was in Grade 5, she was injected with Dengvaxia and had rashes and headaches. I did not know until my child told me because there were children dying from Dengvaxia. My child completed three doses causing her to feel unwell. I did not know it was Dengvaxia; I thought the form I signed was for anti-dengue. My husband and I just prayed because we were afraid. One classmate of hers even died because of Dengvaxia, although there was no proof and autopsy was not done. After that experience, I no longer allow my children to get any of the vaccines. I am also not vaccinated against COVID-19 (B2, 41–50 years old, female, National Capital Region). |
| I6 | Negative experiences with the health system as source of vaccine hesitancy and anti-vaccination sentiments | I am more afraid of vaccines. Once you inject that into your body, you will not be able to reverse its effects or take it out of your system. With COVID, if you are just careful and follow health protocols, and strengthen your immune system, I do not think you will immediately get sick. Compared to the vaccines– we are not sure how safe they are (B2, 41–50 years old, female, National Capital Region). |
| I7 | Vaccines are viewed as unsafe and deadly | So I told my son, "Why do we need a booster"? Because you know, if it's not really 100% [effective], you need a booster after 6 [months] or 1 year. You know what, my son answered me, "You know mom, if you don't die from the first dose, they give you a second dose to make sure that you really die. If you don't die from the second dose, then they give you a booster". It's funny, it's funny (A2, 71–80 years old, female, Misamis Oriental). |
| I8 | Vaccines are viewed as unsafe and deadly | As early as now, we see that after vaccination, people are dying. Although medical professionals do not claim that it is caused by the vaccine. But if we really apply the law of proximate cause. . . Those individuals who went to the site to get vaccinated are healthy, even though they have comorbidities they were okay prior getting the vaccine. Why is it after the vaccination, a person will just suddenly die? [. . .] But then, the doctors would just say that it was not caused by the vaccine. . . that it was because of his comorbidity. Before vaccination, he was [alive] and kicking. But after vaccination, this senior citizen in our place just died. So this is what we are saying. If the vaccine is really a guarantee to solve this problem, then it should not cause mortality (A1, 51–60 years old, female, Misamis Oriental). |
| I9 | Vaccines are viewed as unsafe and deadly | I do not want mRNA vaccines. Once you play around with RNA, you just don't know [what happens after]. The technology is not yet mature (A2, 71–80 years old, male, Laguna). |
| I10 | Vaccines are viewed as unnecessary and insufficient to prevent disease | The elderly are just at home. In other words, they [we] do not go out and interact with people. The vaccine should just be given to the frontliners and those working in the health care sector (A2, 61–70 years old, female, Camarines Norte). |

(*Continued*)

**Table 3.** (Continued)

| Quote ID | Theme | Illustrative quote |
|---|---|---|
| I11 | Vaccines are viewed as unnecessary and insufficient to prevent disease | The elderly and those with comorbidities–they need the vaccine more than I do. In my experience of getting COVID, I only had mild disease. I know that my body can survive. But how about them? How will they survive? (A4, 21–30 years old, female, National Capital Region). |
| I12 | Vaccines are viewed as unnecessary and insufficient to prevent disease | Just because you are vaccinated, it does not mean you are safe from COVID-19. So why should you get the vaccine if you are still not safe in the end? My question is, what are the vaccines for then? (B2, 41–50 years old, female, National Capital Region). |
| I13 | Skepticism towards vaccine incentives | If the vaccine is really that good, then people should be fighting each other to get it. But how come the government has to give you an incentive to get the vaccine? [. . .] If it's really that good, why the incentive? If it's really that good. That's why it bothers me. [. . .] If it is for your protection, if it is for your health, we do not need that [incentive] (A2, 71–80 years old, female, Misamis Oriental). |
| I14 | Use of vaccines not fully approved by the Food Drug and Administration | It [the vaccine] needs to under a thorough process or take many years to have enough clinical studies that can validate the results or so we can see the adverse reactions in the human body. I don't think it's [the development process] this easy that in just a matter of months, we can already use it, right? I don't think it's this easy for them to say that the vaccine is effective to combat the virus (A1, 51–60 years old, female, Misamis Oriental). |
| I15 | Use of vaccines not fully approved by the Food Drug and Administration | I was thinking, maybe this is just an experiment or study. As in the dengue vaccine, they were just studying it and injected it among children, and then found it wasn't safe. I am thinking the same for COVID-19 vaccines (B2, 41–50 years old, female, National Capital Region). |
| I16 | Low health literacy and lack of critical skills to evaluate health information | I noticed educating people is lacking. For example, the father of our house help does not want to get the vaccine because of Dengvaxia. The people we know who belong to lower income groups always say, Dengvaxia. It is frustrating because people are not well-informed about vaccines (A4, 31–40 years old, male, Rizal). |
| I17 | Low health literacy and lack of critical skills to evaluate health information | What people know is superficial and information is not thoroughly discussed in social media or even in informercials of local governments. So what happens sometimes is, they do get side effects like fever, chills during the first vaccination. Then they do not return for the second, which is a wasted opportunity (A1, 51–60 years old, female, Rizal). |
| I18 | Low health literacy and lack of critical skills to evaluate health information | They say the vaccines change your DNA. I don't know. Actually, I don't know what to believe in. If I'm being honest, I don't know what to believe in (A2, 61–70 years old, female, Camarines Norte). |
| I19 | Religious beliefs do not support vaccines | God created natural antibodies, but these will be replaced by man-made vaccines. Nothing can replace what God has created, which may be the reason for the side effects and deaths (A1, 51–60 years old, female, Misamis Oriental). |

administration of the vaccine are caused by the vaccine; however, medical doctors and hospitals report the death as being caused by underlying conditions such as comorbidities (**Quotes I7-I8, Table 3**). Some participants also believed the circulating theory that the life span of those who are vaccinated is shortened and they only have two to three years to live: "*you are healthy but because of the vaccine, you suddenly die*". In addition to the belief that vaccines cause death or shorten an individual's life span, participants also had doubts about the COVID-19 vaccines particularly the mRNA vaccines that use a relatively new technology (**Quote I9, Table 3**). These concerns about the safety profile of vaccines either caused delays in vaccine acceptance and uptake or refusals. The reverse was reported among most of the health workers and other participants who viewed vaccines positively. They believed that the vaccine protects them from severe illness, hospitalisation, and death, and that vaccines only have minimal risk.

**Vaccines are viewed as unnecessary and insufficient to prevent disease.** Vaccines were viewed as unnecessary by some participants, especially those in older age groups who are not allowed to go out (**Quote I10, Table 3**). Those in lower priority groups felt that others needed the vaccine more than them. Younger participants shared that they were COVID-19 survivors even without the vaccine; but those at high risk especially the elderly and persons with comorbidities will need the vaccine to protect them (**Quote I11, Table 3**). The participants also viewed vaccines as insufficient–they expected that getting vaccinated means no longer needing other public health interventions but were disappointed to learn that vaccines are only one part of the solution. Participants therefore questioned the need for the vaccines given the information they have read and/or watched about still being at risk of getting infected despite being vaccinated (**Quote I12, Table 3**). The lack of clarity in the role of the vaccines has negatively influenced people's decisions on getting the vaccine.

**Skepticism towards vaccine incentives.** Vaccine incentives in the country, such as promotions and offers for those vaccinated, created skepticism among some of the participants. These incentives 'bothered' participants and raised questions about the role of vaccines and the intentions of the government. As a result, these incentives 'disincentivised' participants from getting the vaccine as participants felt being forced to take it (**Quote I13, Table 3**).

**Use of vaccines not fully approved by the Food and Drug Administration (FDA).** Participants viewed decisions to vaccinate individuals as 'rash' and expressed concerns about vaccines not yet being fully approved by the Food and Drug Administration (FDA). Some also shared concerns about the rapid development of vaccines compared to other vaccines that took decades to develop (**Quote I14, Table 3**). Participants felt that they were being experimented on using an unproven vaccine, relating this with the dengue vaccine controversy (**Quote I15, Table 3**). This caused delay or refusal in getting the vaccines when it was offered to them.

**Low health literacy and lack of critical skills to evaluate health information.** Health literacy or how people acquire, evaluate, and apply health information to inform their decisions, including getting the vaccine, is an important but underestimated tool to combat misinformation. Participants shared that Filipinos seemed to know a lot about vaccines, but only superficially. They shared that those among low-resource communities and older population groups were especially vulnerable to misinformation (**Quote I16, Table 3**). This lack of awareness and critical skills to evaluate information, together with the rapid spread of misinformation, influences people's decisions to get their first dose, to return to their second and get fully vaccinated (**Quote I17, Table 3**). There were also several participants who shared that they were confused with the contradictory information they were reading and hearing (**Quotes I18, Table 3**).

**Religious beliefs do not support vaccines.** 'Antichrist'–this was how one participant described the vaccines against COVID-19. Another participant shared concerns about the

vaccines and how they would replace antibodies created by God (**Quote I19, Table 3**). She mentioned that these vaccines have active chemicals that are causing unintended side effects and deaths.

## Interpersonal barriers

**Family influence and opposition to vaccines.** Participants recognised the influence of their family on their health decisions including getting vaccinated. One participant who was opposed to COVID-19 vaccines shared that everyone in their family was unvaccinated because they believed her (A1, 51–60 years old, female, Misamis Oriental). Similarly, a mother who had a negative experience related to the dengue vaccine that was administered to her child, refused to have herself and her family vaccinated against COVID-19 (B2, 41–50 years old, female, National Capital Region).

**Misinformation spread by networks.** Rumours and misinformation about COVID-19 vaccines are easily spread by networks, whether by word of mouth or through social media. A participant said her "*eyes have been opened only now because of YouTube*" (A2, 61–70 years old, female, Camarines Norte). Participants believed that this affected vaccine uptake, especially among individuals who do not have the opportunity to receive accurate information from official sources including the Department of Health (**Quote IC1, Table 4**).

**Perceived conflicts of interest of health professionals.** Participants viewed key figures in the response to the pandemic as having conflicts of interests. This perception of having 'hidden agenda' created mistrust in the information provided health professionals, health organisations, and other figures and institutions. These conflicts of interest, whether financial or non-financial, subject evidence and data to bias especially if there are undesired adverse effects to the treatment or vaccine (**Quote IC2, Table 4**).

## Structural barriers: Health systems and service delivery

**Inadequate supply of vaccines.** Observations of participants regarding supply of vaccines varied according to location and membership to the vaccine priority groups. Participants, especially those from cities and provinces outside of metropolitan areas, reported that the supply of vaccines was insufficient to meet the demands and needs of the communities (**Quote S-HS1, Table 5**). However, even within highly urbanised areas, participants shared that there were those who did not get their second doses on time because no vaccines arrived (**Quote**

**Table 4. Illustrative quotes for interpersonal and community barriers.**

| Quote ID | Theme | Illustrative quote |
|---|---|---|
| IC1 | Misinformation spread by networks | People in the remote areas, especially the middle-aged and senior citizens, are apprehensive because they heard from other friends that vaccines may cause damage to their health. [. . .] They said that when you are vaccinated you are given only two years to live and that vaccines contain metals [. . .] so a new generation will come out (A2, 61–70 years old, female, Bukidnon). |
| IC2 | Perceived conflicts of interest of health professionals | In some way, [name of health professional redacted] is funded by some drug companies. Once you are involved in these drug firms, being objective becomes difficult. It is important to be objective, or else you will bias your findings. Although not directly forced [by the drug company], but you have information that you know of but choose to withhold. You will just forget about it, especially if there are unwanted adverse effects [to the treatment or vaccine] (A2, 71–80 years old, male, Laguna). |

**Table 5. Illustrative quotes for structural barriers relating to health systems and service delivery.**

| Quote ID | Theme | Illustrative quote |
|---|---|---|
| S-HS1 | Inadequate supply of vaccines | My [senior citizen, A2] parents in Abra were not able to get their vaccine yet because of inadequate supply of vaccines. I think the vaccination rollout is not yet fully implemented there (A4, 31–40 years old, male, National Capital Region with parents in Abra, Benguet). |
| S-HS2 | Inadequate supply of vaccines | Others are not yet fully vaccinated because they could not get their second dose yet (A4, 31–40 years old, male, National Capital Region with parents in Abra, Benguet). |
| S-HS3 | Inadequate supply of vaccines | Many individuals are not yet vaccinated even in [the first three priority groups] A1 to A3, especially for A2 [senior citizens] and A3 [persons with comorbidities] because we do not have enough supply of vaccines. This is one of the bottlenecks–inadequate supply of vaccines. [. . .] My brother is classified as A3 but he could not get it because there were no more vaccines (A1, 21–30 years old, male, Albay) |
| S-HS4 | Perceived inefficiencies of the vaccination system | This [issue] was when the cases were increasing. We knew the vaccines were going to be needed but the government was late in procuring vaccines. Now the frustration is with the rollout. It is slow. I think we're now at 9%, 10% population [that is vaccinated]. It's going to be a long way, long way to 70% [to reach herd immunity] (A1, 51–60 years old, male, Laguna). |
| S-HS5 | Perceived inefficiencies of the vaccination system | I have a cousin who registered three weeks ago because he is part of the A4 category. Until now, he still did not get a schedule [from the local government]. So what I did, I registered him here at the [health institute]. This week, just this Thursday, he already has a schedule. I just registered him last week (A1, 41–50 years old, male, Laguna). |
| S-HS6 | Perceived inefficiencies of the vaccination system | "*Nadidismaya*" [or disappointed] because on the day of vaccination, there is a two-hour seminar about COVID and vaccines [. . .] There were people leaving the vaccination site because they found the two-hour seminar long, and they were afraid of crowding in one area (A2, 61–70 years old, male, Nueva Vizcaya). |
| S-HS7 | Perceived inefficiencies of the vaccination system | I believe there's this glitch in the registration system of the city. There's a bug in the system that significantly slows down the vaccine rollout (B1, 21–30 years old, male, National Capital Region). |
| S-HS8 | Perceived inefficiencies of the vaccination system | The system is not centralised. For example, you were already vaccinated in one site. But you are also in the list of another site. I think there should be a feedback mechanism, "I am vaccinated already, you may remove me from the list". So that others waiting can get the slot (A4, 31–40 years old, male, Rizal). |
| S-HS9 | View that the vaccination system is inflexible and excludes vulnerable and marginalised populations | I got my vaccine ahead of my parents. Because I would always wait for posts from the local government on Facebook. At the time they announced that registration was open, I registered myself immediately. But my parents, they are not inclined to technology so they would just wait for guidance (C, 21–30 years old, male, Cebu). |

(*Continued*)

**Table 5.** (Continued)

| Quote ID | Theme | Illustrative quote |
|---|---|---|
| S-HS10 | View that the vaccination system is inflexible and excludes vulnerable and marginalised populations | The process takes very long. First, there are so many steps needed during vaccination. They take the [person's] information like name–they will write your name. Next step, this is repeated, they ask for your name again. Then they will ask, "do you have allergies"? There are so many steps that could have been placed in one station [at the vaccination site]. And another thing, the lines. Everyone who will be vaccinated is seated. Once someone is done, everyone stands up and transfers to the next chair. If you are old, can you imagine standing up, sitting down, standing up, sitting down? It is painful on the knees. They should just call everyone in a row, or they should just call people at a time, with everyone else staying in their seats. They have so many steps just asking for the name, these steps take so much time. One and a half hours. For seniors, this is difficult especially because everyone lines up [and there is no special lane for seniors] (A2, 71–80 years old, male, Laguna). |
| S-HS11 | View that the vaccination system is inflexible and excludes vulnerable and marginalised populations | My grandmother [A2 priority] is not yet vaccinated. She shared with me that you need to line up at the basketball court. My grandmother has difficulties walking so she cannot go there to line up. She also has difficulty breathing and there are times she needs oxygen when going out. So we [family] have this fear that if we bring her there, instead of it [getting the vaccine] being a good thing, she might get infected. One other reason is transportation so her concern is if the vaccination schedule is announced, "How will I go there?". The local government does not have home vaccination or services that bring the vaccines to individuals' homes. But she wants to get the vaccine (A4, 21–30 years old, female, Cavite) |
| S-HS12 | Logistical challenges | I think logistics is also delaying the vaccination campaign. For example, Pfizer has special requirements for storage (A1, 51–60 years old, male, Laguna). |
| S-HS13 | Logistical challenges | Why do I prefer Moderna? Although Pfizer is more effective, but its handling is difficult. It requires cold storage. [. . .] For this reason, my number one preferred brand is Moderna (A4, 31–40 years old, male, Rizal). |
| S-HS14 | Health professionals seen as amplifiers of misinformation | In addition to Ivermectin, vaccines are another debate within the medical community. I even have a classmate [in medical school] who is an anti-vaxxer. I said, "Let's wait" [for the evidence]. Because others were already fighting (A1, 51–60 years old, male, Laguna). |
| S-HS15 | Health professionals seen as amplifiers of misinformation | There are actually doctors who are anti-vaxxers. There is this specific doctor who had a talk with a public radio station. Previously, she would not give vaccines, according to my classmates [in medical school] working with her because she is doing private practice. She had a pregnant patient who was referred to her. She would convince the mother not to have the child vaccinated (A1, 21–30 years old, male, Iloilo). |

(*Continued*)

**Table 5.** (Continued)

| Quote ID | Theme | Illustrative quote |
|---|---|---|
| S-HS16 | Pandemic response deemed as ineffective affects trust in health institutions | They're [institutions] not doing their job they're supposed to do. That's not a political statement, that is the comment of the people on the ground. Us, we in the masses. . . They are just giving us lies and inciting fear, and misleading [us] (A3, 51–60 years old, male, Nueva Vizcaya). |
| S-HS17 | Pandemic response deemed as ineffective affects trust in health institutions | The Department of Health has daily updates, right? It's unfortunate that a good number of reacts [on Facebook] are 'haha' reacts. Well, I would understand that because it's been more than a year and we are still at 5,000 to 6,000 cases per day. I would understand the angry reacts. They would say why is the system not better, or why are we not getting vaccines right away? It indicates that people are willing to do their part in stemming the spread of the virus. But with 'haha' reacts, what's funny? What's posted there are the number of people who died that day. These people, come on [in disbelief] (B1, 21–30 years old, male, National Capital Region). |
| S-HS18 | Pandemic response deemed as ineffective affects trust in health institutions | Before, people were worried because of the news circulating in social media about the side effects [of the vaccine]. It's a good thing I'm in the health sector so I know that vaccines are needed. But if I were not a health worker, I would not get vaccinated because I might suddenly die. Or maybe the government is just not telling the truth (A1, 41–50 years old, female, Rizal). |

**S-HS2, Table 5**). Health workers found that vaccines for them were easily accessible, however those in other groups had to wait longer before getting the vaccine (**Quote S-HS3, Table 5**).

**Perceived inefficiencies of the vaccination system.** Participants highlighted issues with the system including the slow rollout of vaccines, long waiting time, inefficient registration systems, and lack of a centralised system. Participants mentioned getting frustrated with the speed at which vaccines are being distributed and administered in the country (**Quote S-HS4, Table 5**). Participants also mentioned issues with the waiting process to get a slot after registration and the waiting time at the day of the vaccination, with some being asked to stay at vaccination sites for two hours to watch a seminar on COVID-19 and vaccines (**Quotes S-HS5-6, Table 5**). There was perceived risk of exposure, which could be lessened if the process was faster and more efficient. There were also glitches in the online registration systems used by local governments that caused additional delays in getting people vaccinated (**Quote S-HS7, Table 5**). Local governments are responsible for the distribution and administration of vaccines among their constituents, and individuals may register with various local governments depending on their place of residence or work. This lack of a centralised system makes it difficult to track who have already been vaccinated and where they have been vaccinated such that those who are still waiting for a slot are unable to secure one (**Quote S-HS8, Table 5**).

**View that the vaccination system is inflexible and excludes vulnerable and marginalised populations.** The current vaccination system of some local governments is viewed as inflexible that excludes vulnerable and marginalised populations. There are individuals who lack access to technology and digital platforms. Especially in rural areas and among the elderly, their exclusion due to access issues is further compounded by their low digital health literacy. These individuals are then unable to register online and get the vaccine (**Quote S-HS9,**

Table 5). While registration is online, even those in older age groups who are part of highly prioritised groups because of their susceptibility to the virus are required to go to the vaccination centre (**Quote S-HS10, Table 5**). Similarly, those belonging to marginalised groups and communities also encounter considerable challenges to getting the vaccine (**Quote S-HS11, Table 5**).

**Logistical challenges.**   A participant recognised that there are also logistical constraints in the distribution of vaccines, in addition to problems with supply. The COVID-19 vaccines have different temperature requirements with some requiring special distribution systems (**S-HS12, Table 5**). These logistical challenges influence the distribution of vaccine brands to areas that have the capability to store them and affect decisions to delay getting the vaccine especially among those who prefer other brands (**S-HS13, Table 5**).

**Health professionals seen as amplifiers of misinformation.**   Misinformation on vaccines and treatment were not only observed within families and social networks, but also within the medical community reported by participants who are health professionals themselves. There have been debates about Ivermectin as treatment for COVID-19, as well as vaccines, which have created factions within the group (**S-HS14, Table 5**). Some of these health professionals who are anti-vaxxers or opposed to vaccines publicly share their views in media and in their practice (**S-HS15, Table 5**). Because of the stature and credibility of health professionals, their views, whether backed by science or not, get amplified in the media and communities.

**Pandemic response deemed as ineffective affects trust in health institutions.**   The response and messaging of health organisations, together with other key figures and institutions in the country, were viewed by participants as ineffective (**S-HS16, Table 5**). As a result, there is declining trust in these organisations with participants doubting information provided, such that Filipinos no longer take the pandemic seriously (**S-HS17, S-HS18, Table 5**). In turn, participants turn to other sources of information that they think are more credible and trustworthy.

## Structural barriers: Media and policies

**Traditional and digital media accelerating the infodemic.**   Information on the virus and vaccines are easily and effectively amplified by the media. With the infodemic (portmanteau of information and epidemic) or the exponential production of information whether scientifically accurate or not, traditional media and digital media become drivers of (mis)information or fear towards vaccines (**Quotes S-MP1-S-MP2, Table 6**). Information that participants were receiving from these sources influenced their health beliefs and vaccine decisions (**Quote S-MP3, Table 6**).

**Perceived poor policy implementation and lack of evidence-based policies contributing to loss of confidence in vaccines and health institutions.**   The government developed the Philippine "National Deployment and Vaccination Plan for COVID-19 Vaccines" that identifies population groups to be prioritised ensure vaccine equity accounting for different risks and needs [36]. This plan also stated that only vaccines granted with emergency use authorisation (EUA) or certificate of product registration (CPR) by the Philippine FDA will be purchased by the government. However, this was reported by participants to be poorly implemented with others using connections also known as '*palakasan*' system to get the vaccine ahead of those in the priority list (**Quote S-MP4, Table 6**). Even within the government, the Presidential Security Group were given vaccines even without EUA and/or CPR registration from the FDA (**Quote S-MP5, Table 6**). In addition, the government purchased vaccines that did not publish their results, and reportedly had lower efficacy rates but more expensive (**Quote S-MP6, Table 6**). As a result, participants felt that the government was 'settling for

**Table 6. Illustrative quotes for structural barriers relating to the media and policies.**

| Quote ID | Theme | Illustrative quote |
|---|---|---|
| S-MP1 | Traditional and digital media accelerating the infodemic | When Sinovac arrived, there were people who refused. People were afraid of what the media was reporting about the vaccine. For example, my secretary refused to get vaccinated. [. . .] Because there was a lot of news about it. I don't know if fake news, but there were fears because it's a new vaccine, it's not been tested (A1, 51–60 years old, male, Laguna). |
| S-MP2 | Traditional and digital media accelerating the infodemic | I got Pfizer from our government, and it was not easy at the start because of the negative stories that I would hear. The hearsays from social media [. . .] like Facebook and Twitter. Because people were reacting already then when the rollout of vaccinations started. There were stories about people who had negative body reactions to the first dose, hearsays from people who would say that the vaccine would shorten your life span to five years? There were stories like that. It got me thinking and it got me asking. When people in our community would ask me if I got vaccinated, I would say "No, I am afraid" (A3, 41–50 years old, female, National Capital Region). |
| S-MP3 | Traditional and digital media accelerating the infodemic | Have you read about the New World Order? I read that this is a 'plandemic' instead of pandemic. In other words, this virus was made in the laboratory in Wuhan, China with the purpose of depopulating the world. [. . .] I now believe that we are in the End Times as mentioned in the Bible. It was only now that my eyes have been opened because of YouTube (A2, 61–70 years old, female, Camarines Norte). |
| S-MP4 | Perceived poor policy implementation and lack of evidence-based policies contributing to loss of confidence in vaccines and health institutions | We cannot avoid it–that those who have connections get the vaccine first. And then they [government] promised us that we frontliners and our family members [extended priority list] will be prioritised for the vaccines. But this is not true. I was not prioritised and this was the same experience for my co-workers (A1, 31–40 years old, female, Pampanga). |
| S-MP5 | Perceived poor policy implementation and lack of evidence-based policies contributing to loss of confidence in vaccines and health institutions | The FDA approved Sinopharm [for compassionate use] even if there were no published trials yet. The vaccination of the Presidential Security Group was illegal. Why were they vaccinated illegally? There are many issues with the government, which make people question the vaccines. So there were many people who hesitated getting the vaccine, and they lost confidence in the available vaccines. We, health workers, had difficulties persuading or convincing people to get vaccinated (A1, 21–30 years old, male, Albay). |
| S-MP6 | Perceived poor policy implementation and lack of evidence-based policies contributing to loss of confidence in vaccines and health institutions | As I have mentioned before, I tend to decide based on what I know and what I have read. Most of the vaccines that the government ordered are Sinovac, which did not undergo phase 3 and peer review. This is the reason why I don't believe in our government. Also, Sinovac is more expensive but has a lower efficacy rate compared to other vaccines which are cheaper but has higher efficacy rate like AstraZeneca. Now ask yourself why would your government prefer a vaccine that is more expensive but with lower efficacy for its constituents if our government applied for loans in international banks? (A4, 21–30 years old, male, National Capital Region) |
| S-MP7 | National and local political issues | My least preferred vaccine brand is Sinovac because of its country of origin. I do not believe in China. Directly, you can put that on record. Because of their products and medicines, and also what they're doing to us with the West Philippine Sea. Those things are also now being considered by people. For me, at least for me. I'm speaking for myself. I don't like what they're doing to us as a country. You can place that on record (A3, 51–60 years old, male, Nueva Vizcaya). |

(*Continued*)

**Table 6.** (Continued)

| Quote ID | Theme | Illustrative quote |
|---|---|---|
| S-MP8 | National and local political issues | I watched a video about a scientist in China who is hiding in Hong Kong. This scientist was revealing the truth about COVID and how it was created. I think this was created by China to take over countries especially with what happened with the West Philippine Sea. They want to make the Philippines a part of their country. [. . .] With vaccines, I think other countries are angry at China because that's where the virus came from. So these countries developed their own vaccines as a defense (B2, 41–50 years old, female, National Capital Region). |
| S-MP9 | National and local political issues | I decided to get Sinovac—I am at risk of getting infected with the virus because I am frequently exposed to people. I also don't think I have the chance to get Pfizer because it is being shipped to Davao [city where the President and his family resides]. I would be lucky to get Pfizer, but I do not have connections with the government (A1, 31–40 years old, female, Pampanga). |

less' and that Filipinos deserved better (A4, 21–30 years old, female, National Capital Region). These issues contributed to declining confidence in vaccines and health institutions, with Filipinos questioning the safety of such vaccines and the implementation of these prioritisation frameworks.

**National and local political issues.** Past and current political issues contributed to refusals to specific vaccine brands. Together with reports of how the virus emerged from Wuhan, China, these triggered skepticism towards vaccines manufactured in their country. Participants mentioned the dispute of the Philippines and China regarding contested territory at the West Philippine Sea (South China Sea) as a reason for not preferring and/or refusing vaccines from their country, even when donations of Sinovac from China were the first vaccines to be available (**Quote S-MP7, Table 6**). This dispute also influenced how participants thought about the origins of the virus and why other countries developed their own vaccines (**Quote S-MP8, Table 6**). Locally, participants viewed politics to have influence on which cities or provinces receive preferred vaccine brands. They mentioned that these 'favored hospitals and provinces' were prioritised, which was perceived as unfair and causing further delays in the vaccination rollout (**Quote S-MP9, Table 6**).

## Discussion

One of the most effective public health strategies, vaccination, has been the focus of false and inaccurate information with rapidly declining rates of acceptance. [37]. In the Philippines, vaccine confidence plummeted after the Dengue vaccine controversy [9, 12, 15–17]. While anti-vaccination views and vaccine hesitancy are not yet the main barrier to vaccination in the Philippines which still struggles with vaccine access and distribution, lessons from other countries indicate that these equally and urgently need to be addressed in addition to other challenges [38]. Our study supports the findings of other published research that report a host of individual, interpersonal, and structural barriers that work individually or collectively against vaccination uptake and reach [29]. Therefore, there is a need for a holistic approach to promote

COVID-19 vaccination that not only addresses barriers at the individual level, but also at the interpersonal and structural levels [38, 39].

Individual perceptions, beliefs, and experiences play a major role on the decision to vaccinate. These are shaped by exposure to (mis)information spread by networks, by key health figures and institutions, and through the media [40–43]. Misinformation regarding vaccines have been present since vaccines were first developed [44–46], but the advent of social media made its propagation much easier [43, 45, 47]. Unique to the Philippine context is vaccine brand hesitancy, specifically towards Chinese manufactured vaccines and mRNA vaccines. This is caused in part by lack of transparency and scientific information, and spread through networks and the media. Further aggravating the issue is how some people attempt to correct misconceptions in a way that alienates people instead of addressing misinformation. People involved in vaccine promotion activities, especially primary care providers, may need to be trained on how to engage with vocal vaccine deniers and promote vaccination. The World Health Organization document outlining how to respond to vaccine misinformation would be an important resource in such an endeavour [48]. Celebrities and social media influencers may also play a role in promoting vaccination [41], but it is essential that they disclose conflicts of interest to develop trust with their audience. The media also needs to be trained on how to present news regarding adverse effects following immunsation, and regarding COVID-19 in general, so as not to create unnecessary panic and dissuade people from getting vaccinated. A study reported that there may be a need to use first-person, people-centred narratives to prevent 'psychic numbing' and give faces to numbers [49]. In all these, it is vital to engage with the public, especially those who are vaccine hesitant, in order to promote vaccination using language that is inclusive and applicable to their context [48].

The health system and one's interactions with it also contribute to one's decision to get vaccinated. As in this study, trust in the health system has been found to be a major factor in getting COVID-19 vaccine [41, 50]. The Philippine government has instituted several health system confidence-building policies. The recent COVID-19 Vaccination Program Act stipulates the provision of free COVID-19 vaccines to all Filipinos and the establishment of an indemnification fund for people who could possibly develop adverse effects following immunisation [51]. Perceptions of 'palakasan' (i.e., use of political connections), stemming from instances during the course of the pandemic where powerful individuals seem to be above the law [52], contribute to vaccine hesitancy and poor uptake of vaccines. These negative impacts are further compounded by the highly politicised Dengvaxia controversy where individuals, especially parents of school-age children, felt that health institutions and governments were experimenting on them [9, 12] with our participants relating the COVID-19 vaccine 'experiment' with the dengue vaccine. In addition, inadequate supply, logistical challenges, and perceptions about the inefficiency and inflexibility of the system negatively impact vaccination rates in the country. As of 16 September 2021, only 3 in 10 Filipinos received one dose with significant differences between population groups: almost all frontline and health workers have been vaccinated while only 2 in 5 elderly Filipinos received their first dose [21]. Those in the third priority group have higher rates than the elderly population group, which were offered the vaccines earlier. Apart from individual reasons, marginalised and vulnerable groups such as the elderly have reported not being able to get their vaccine due to lack of home vaccination services and guidance in using online registration systems. The system will need to consider needs of all population groups to improve vaccination uptake. In all these, trust in the health system needs to be maintained, while disregarding regulations and policies in place can erode trust in the vaccination process.

In the Philippines, the national government has the responsibility to procure, allocate, and distribute the vaccines to the different provinces and municipalities, but it is the local

government that is responsible for last-mile transport and actual inoculation. This results in wide variations in client registration and procedures between different localities. This underlines the need to identify best practices in vaccine rollout systems to implement a system that is efficient and inclusive to ensure that access to technology and mobility will not be barriers to vaccination.

There are a number of limitations that need to be considered when interpreting our findings. First, we were not able to have representation from the A5 priority group (indigent population). While we initially were able to get a participant from this group based on the survey response, we later found during the interview that this individual belonged to a different vaccination priority classification. This may point to issues with online data collection where researchers are unable to reach individuals from low-resource households. Second, there may be social desirability bias because we were unable to ensure if the respondent had other people with them that may have caused a change in their responses. Additionally, we did not disclose any political affiliations and interests, but participants may have been cautious in mentioning negative experiences related to vaccination. Participants may also have chosen more positive responses considering our background as health researchers. However, we emphasised that they will remain anonymous and their data treated with utmost confidentiality. Lastly, factors influencing COVID-19 vaccination uptake is context-specific, and this paper does not aim to represent all situations and circumstances.

## Conclusion

Challenges to COVID-19 vaccination may be individual, interpersonal, and/or structural, which interact to influence decisions. Individual perceptions play a major role in the decision to vaccinate, and such perceptions are shaped by exposure to (mis)information amplified by the media, the community, and the health system. In the Philippines, vaccine brand hesitancy and misinformation are prevalent due to their rapid spread through social media and sensationalism in traditional media. Information on the effectiveness of safety of vaccines regardless of brand needs to be communicated to the public to increase COVID-19 vaccine confidence. At the interpersonal level, exposure to networks and health workers who are opposed to vaccines heightens public skepticism of vaccination. Structural barriers including political issues and poor implementation further contribute to vaccine refusals. The ongoing infodemic and anti-vaccination sentiments operating at all three levels (individual, interpersonal, structural) require empowering individuals to evaluate health information, and therefore health literacy becomes a critical tool to combat misinformation. Families and peers also need to be involved in these discussions as they influence vaccine uptake. Individuals engaged in vaccine promotion activities may need to be retrained on how to engage with vocal vaccine deniers in public. Given the involvement of traditional media, trainings on public health and science communication may be helpful in reporting vaccination-related news. Public figures need to disclose conflicts of interests and be transparent to the public, laying out the risks and benefits of vaccines. Laws should be well-implemented and equally implemented regardless of socioeconomic class or social position to encourage trust in the health care system and in vaccination initiatives. There is also a need to study best practices in vaccine rollout to implement systems that are efficient and inclusive so that we can vaccinate as many people against COVID-19 as quickly and as inclusively as possible: provide technological support particularly among older populations and allow flexible options for receiving the vaccine such as home vaccination. Given resource limitations, the vaccination rollout could also be improved by increasing the role of the private sector in the rollout and administration of the vaccine. The government and health organisations will need to connect with individuals, communities, and other

institutions, including those who are against vaccines or hesitant towards vaccines, to co-create effective and sustainable solutions.

## Acknowledgments

We would like to thank Michelle Edillon, Kriselle Abcede, Ryan Molen, and Josef Bondoc for their invaluable support to this project. We provide credit to BioRender.com for the figures illustrated in this paper. Finally, we are grateful to our participants who generously shared their stories with us.

## Author Contributions

**Conceptualization:** Arianna Maever L. Amit, Veincent Christian F. Pepito, Lourdes Sumpaico-Tanchanco, Manuel M. Dayrit.

**Data curation:** Arianna Maever L. Amit, Veincent Christian F. Pepito.

**Formal analysis:** Arianna Maever L. Amit, Veincent Christian F. Pepito.

**Funding acquisition:** Lourdes Sumpaico-Tanchanco, Manuel M. Dayrit.

**Investigation:** Arianna Maever L. Amit, Veincent Christian F. Pepito, Lourdes Sumpaico-Tanchanco, Manuel M. Dayrit.

**Methodology:** Arianna Maever L. Amit, Veincent Christian F. Pepito, Lourdes Sumpaico-Tanchanco, Manuel M. Dayrit.

**Project administration:** Arianna Maever L. Amit, Lourdes Sumpaico-Tanchanco.

**Resources:** Lourdes Sumpaico-Tanchanco.

**Supervision:** Lourdes Sumpaico-Tanchanco, Manuel M. Dayrit.

**Validation:** Veincent Christian F. Pepito, Lourdes Sumpaico-Tanchanco, Manuel M. Dayrit.

**Visualization:** Arianna Maever L. Amit.

**Writing – original draft:** Arianna Maever L. Amit, Veincent Christian F. Pepito.

**Writing – review & editing:** Arianna Maever L. Amit, Veincent Christian F. Pepito, Lourdes Sumpaico-Tanchanco, Manuel M. Dayrit.

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
