## [Decision Letter · Decision Letter 0]

3 Dec 2021

PGPH-D-21-00884

COVID-19 vaccine brand hesitancy and other challenges to vaccination in the Philippines

Dear Dr. Amit,

Thank you for submitting your manuscript to PLOS Global Public Health. After careful consideration, we feel that it has merit but does not fully meet PLOS Global Public Health’s publication criteria as it currently stands. Therefore, we invite you to submit a revised version of the manuscript that addresses the points raised during the review process.

We look forward to receiving your revised manuscript.

Kind regards,

Dione Benjumea-Bedoya, Ph.D

Guest Editor

Journal Requirements:

2. Please update the completed 'Competing Interests' statement, including any COIs declared by your co-authors. If you have no competing interests to declare, please state "The authors have declared that no competing interests exist". Otherwise please declare all competing interests beginning with the statement "I have read the journal's policy and the authors of this manuscript have the following competing interests:

3. We see that your study includes live participants, but you have not included an Ethics Statement. Please update your manuscript file to include an Ethics Statement subsection to your Materials and Methods section. It should include:

iii) (for human participants or donors) - A statement that formal consent was obtained (must state whether verbal/written) OR the reason consent was not obtained (e.g. anonymity)

4. Please provide  separate figure files in .tif or .eps format only and remove any figures embedded in your manuscript file.  Please ensure that all files are under our size limit of 20MB.  

For more information about how to convert your figure files please see our guidelines: Once you've converted your files to .tif or .eps, please also make sure that your figures meet our format requirements

5. Please amend your detailed Financial Disclosure statement. This is published with the article, therefore should be completed in full sentences and contain the exact wording you wish to be published.

ii). State the initials, alongside each funding source, of each author to receive each grant.

iii). State what role the funders took in the study. If the funders had no role in your study, please state: “The funders had no role in study design, data collection and analysis, decision to publish, or preparation of the manuscript.”

iv). If any authors received a salary from any of your funders, please state which authors and which funders.

Additional Editor Comments (if provided):

The authors presented a complete and very good quality work, which contributes to broadening the knowledge about COVID-19 vaccine hesitancy. The reviewers agree on the relevance of this manuscript as well as its good quality, and made some recommendations that are presented below and in the attached file, so that the manuscript becomes suitable to be considered for publishing. 

Reviewers' comments:

Reviewer's Responses to Questions

**Comments to the Author**

1. Does this manuscript meet PLOS Global Public Health’s publication criteria? Is the manuscript technically sound, and do the data support the conclusions? The manuscript must describe methodologically and ethically rigorous research with conclusions that are appropriately drawn based on the data presented.

Reviewer #1: Yes

Reviewer #2: Yes

2. Has the statistical analysis been performed appropriately and rigorously?

Reviewer #1: Yes

Reviewer #2: Yes

3. Have the authors made all data underlying the findings in their manuscript fully available (please refer to the Data Availability Statement at the start of the manuscript PDF file)?

Reviewer #1: Yes

Reviewer #2: Yes

4. Is the manuscript presented in an intelligible fashion and written in standard English?

Reviewer #1: Yes

Reviewer #2: Yes

5. Review Comments to the Author

Reviewer #1: The article presented is of excellent quality, urgent and necessary to understand the complexity in the face of hesitations to vaccinate for Covid-19.

The authors designed a coherent qualitative study that, through semi-structured interviews, content analysis and grounded theory, managed to reach pertinent and coherent conclusions regarding the individual, interpersonal and structural reasons for not accepting vaccination in the Philippines.

The authors are very organized in their analysis strategy and presentation of results that allow coherently linking the categories with the testimonies that support them, organized in the three main dimensions of reasons: individual, interpersonal and structural.

Promoting this multilevel understanding should guide health authorities to guide multilevel actions in the face of the complexity of hesitations regarding vaccination for covid-19, not only in the Philippines, but also in similar social and cultural contexts.

Reviewer #2: It is a very good job, the situation described in the Philippines is similar to other places in the world.

I consider that you should make clear how you avoided the loss of information when translating from Filipino and Cebuano to English.

You should revise reference number 2, it is not in accordance with the standard.

6. PLOS authors have the option to publish the peer review history of their article (what does this mean?). If published, this will include your full peer review and any attached files.

**Do you want your identity to be public for this peer review?** For information about this choice, including consent withdrawal, please see our Privacy Policy.

Reviewer #1: **Yes: **Samuel Arias-Valencia

Reviewer #2: No

---

## [Editor Report · Decision Letter 1]

22 Dec 2021

COVID-19 vaccine brand hesitancy and other challenges to vaccination in the Philippines

PGPH-D-21-00884R1

Dear Dr. Amit,

We're pleased to inform you that your manuscript has been judged scientifically suitable for publication and will be formally accepted for publication once it meets all outstanding technical requirements.

Within one week, you'll receive an e-mail detailing the required amendments. When these have been addressed, you'll receive a formal acceptance letter and your manuscript will be scheduled for publication.

An invoice for payment will follow shortly after the formal acceptance. To ensure an efficient process, please log into Editorial Manager at https://www.editorialmanager.com/pgph/ click the 'Update My Information' link at the top of the page, and double check that your user information is up-to-date. If you have any billing related questions, please contact our Author Billing department directly at authorbilling@plos.org.

Kind regards,

Dione Benjumea-Bedoya, Ph.D

Guest Editor

Additional Editor Comments (optional):

The authors accepted and reviewed all the reviewers' recommendations, which they included in the manuscript. They also made some changes to the text improving its quality.